# Retention and predictors of attrition among patients who started antiretroviral therapy in Zimbabwe's national antiretroviral therapy programme between 2012 and 2015

**Richard Makurumidze**[1,2,3‡]*, **Tsitsi Mutasa-Apollo**[4‡], **Tom Decroo**[2,5], **Regis C. Choto**[4], **Kudakwashe C. Takarinda**[4,6], **Janet Dzangare**[4], **Lutgarde Lynen**[2], **Wim Van Damme**[2,3], **James Hakim**[1], **Tapuwa Magure**[7], **Owen Mugurungi**[4], **Simbarashe Rusakaniko**[1]

**1** College of Health Sciences, University of Zimbabwe, Harare, Zimbabwe, **2** Institute of Tropical Medicine, Antwerp, Belgium, **3** Gerontology, Faculty of Medicine & Pharmacy, Free University of Brussels (VUB), Brussels, Belgium, **4** AIDS & TB Unit, Ministry of Health & Child Care, Harare, Zimbabwe, **5** Research Foundation of Flanders, Brussels, Belgium, **6** International Union Against Tuberculosis and Lung Disease, Paris, France, **7** National AIDS Council, Harare, Zimbabwe

‡ RM and TMA are joint first authors on this work.
* rmakurumidze@ext.itg.be

**Data Availability Statement:** The study was conducted with routinely collected data of the

## Abstract

### Background

The last evaluation to assess outcomes for patients receiving antiretroviral therapy (ART) through the Zimbabwe public sector was conducted in 2011, covering the 2007–2010 cohorts. The reported retention at 6, 12, 24 and 36 months were 90.7%, 78.1%, 68.8% and 64.4%, respectively. We report findings of a follow-up evaluation for the 2012–2015 cohorts to assess the implementation and impact of recommendations from this prior evaluation.

### Methods

A nationwide retrospective study was conducted in 2016. Multi-stage proportional sampling was used to select health facilities and study participants records. The data extracted from patient manual records included demographic, baseline clinical characteristics and patient outcomes (active on treatment, died, transferred out, stopped ART and lost to follow-up (LTFU)) at 6, 12, 24 and 36 months. The data were analysed using Stata/IC 14.2. Retention was estimated using survival analysis. The predictors associated with attrition were determined using a multivariate Cox regression model.

### Results

A total of 3,810 participants were recruited in the study. The median age in years was 35 (IQR: 28–42). Overall, retention increased to 92.4% (p-value = 0.060), 86.5% (p-value<0.001), 79.2% (p-value<0.001) and 74.4% (p-value<0.001) at 6, 12, 24 and 36 months respectively. LTFU accounted for 98% of attrition. Being an adolescent or a young adult (15–24 years) (vs adult;1.41; 95% CI:1.14–1.74), children (<15years) (vs adults; aHR

Zimbabwe National ART Programme and anonymized individual patient level data. Permission to conduct the study and ethical clearance were obtained from the Ministry of Health & Child Care and the Medical Research Council of Zimbabwe. Permission was also sought to disseminate the results in relevant scientific forums. However, the data which was used to conduct is not available on the public domain and anyone interested in using the data for scientific purpose is free to request permission from the Director of the AIDS and TB Program, AIDS and TB Unit, Ministry of Health and Child Care, Government of Zimbabwe, 2nd Floor, Mukwati Building, Harare, Zimbabwe. E-mail: atp. director@ymail.com.

**Funding:** The study was supported with funding from the Global Fund for AIDS, Tuberculosis, and Malaria (GFATM). The funders had no role in study design, data collection and analysis, decision to publish, or preparation of the manuscript. Richard Makurumidze receives a PhD scholarship grant from the Institute of Tropical Medicine, funded by the Belgian Development Cooperation.

**Competing interests:** The authors have declared that no competing interests exist.

0.64; 95% CI:0.46–0.91), receiving care at primary health care facility (vs central and provincial facility; aHR 1.23; 95% CI:1.01–1.49), having initiated ART between 2014–2015 (vs 2012–2013; aHR1.45; 95%CI:1.24–1.69), having WHO Stage IV (vs Stage I-III; aHR2.06; 95%CI:1.51–2.81) and impaired functional status (vs normal status; aHR1.25; 95%CI:1.04–1.49) predicted attrition.

## Conclusion

The overall retention was higher in comparison to the previous 2007–2010 evaluation. Further studies to understand why attrition was found to be higher at primary health care facilities are warranted. Implementation of strategies for managing patients with advanced HIV disease, differentiated care for adolescents and young adults and tracking of LTFU clients should be prioritised to further improve retention.

## Introduction

Globally by the end of 2018, there were 37.9 million [32.7 million–44.0 million] people living with HIV (PLHIV) with 61% of these residing in Eastern and Southern Africa (ESA) [1]. Over the past two to three decades, investments in the global HIV response have achieved unprecedented results with the number of new HIV infections significantly reduced from2.9 million [2.3 million–3.8 million] in 1997 to 1.7 million [1.6 million–2.3 million] new infections by 2018and 23.3million PLHIV put on treatment globally [1]. Between 2010 and 2018 new HIV infections and deaths decreased by 16% and 33%, respectively [2]. Despite these remarkable achievements, patient attrition and losses to follow-up (LTFU) still remain legitimate threats to the long-term success of antiretroviral therapy (ART) scale up [3].

PLHIV on ART who are not retained in care are at increased risk of developing drug resistance and dying [4]. Based on a systematic review of several studies that have been conducted in low resource settings, key predictors of high attrition include patients with advanced HIV-disease progression [marked by body mass index (BMI) <18 kg/m$^2$, baseline CD4 counts <200 cells/mL, World Health Organisation (WHO) Stage—III and IV, poorer level of functionality], male sex, younger age and having lower levels of education [5]. Early ART initiation as measured by shorter time duration between HIV testing and ART initiation has been shown to reduce risk of attrition [6,7]. Other studies have however shown that those who initiate ART at a higher baseline CD4 may also be prone to attrition [8,9].

Numerous studies assessed attrition and the effect of different initiatives on treatment outcomes at selected health facilities in resource-limited settings [7,8,10–12]. They mainly reflect the experience of academic, standalone, donor supported or private institutions that generally have better data collection systems, are well financed and have better human resources for health, which may not be generalizable. In scenarios where outcomes for routine programmes have been reported, evaluations have often been regional, limited to a few facilities or targeting only a specific subpopulation of the HIV cohorts (children, adolescents or adults) [6,13–17]. There is paucity of data demonstrating treatment outcomes and impact of ART at a national level in many resource-limited settings. Periodic national treatment outcome evaluations provide information needed for targeted interventions that will allow national HIV programmes to achieve the ambitious UNAIDS 3$^{rd}$ 90 by 2020.

Zimbabwe started rolling out ART in the public health system in 2004 in 5 pilot facilities. Since then, there has been significant scale-up with more than 1500 health institutions offering ART by the end of 2017 [18]. This rapid scale-up has been mostly attributed to the rapid adaptation of the WHO HIV guidelines and the widespread decentralization of comprehensive HIV services which was supported by health care worker task-shifting policies and significant investments in training, supportive supervision and clinical mentoring [19]. Funding from the Government of Zimbabwe, Global Fund for AIDS, Tuberculosis, and Malaria (GFATM), the United States of America President's Emergency Plan for AIDS Relief (PEPFAR) and other donors supported the scale-up the program.

An evaluation of the Zimbabwe national ART programme, which only included adults ($>$ 15 years) living with HIV started on ART between 2007–2010, reported 90.7%, 78.1%, 68.8% and 64.4% retention at 6, 12, 24 and 36 months, respectively [17]. These findings mirrored treatment outcomes in other parts of sub-Saharan Africa and recommendations given included strengthening earlier diagnosis and linkages to treatment; further decentralization of comprehensive HIV services to improve ART coverage and adaptation of innovative strategies aiming at improving patient retention (adherence clubs, food supplementation and mobile short messages service (SMS) reminders) [17].

In our study, we report results of a 2016 follow-up evaluation of treatment outcomes of the Zimbabwe National ART Programme, conducted among individuals who started ART between 2012–2015, prior to the implementation of the HIV 'Treat All' policy. This nationwide follow-up evaluation therefore assessed the impact of the interventions implemented since the first evaluation. The assessment was expanded to include children and adolescents. We expected an increase in retention on ART following interventions recommended after the previous evaluation. The aim was to estimate retention, compare it with the previous evaluation, and explore predictors of attrition (either lost to follow-up or died).

## Material & methods

### Study design

A retrospective cohort analysis was undertaken among children, adolescents and adults living with HIV who started ART in Zimbabwe between October 2012 and January 2015.

### Study setting

The study was conducted at selected public health institutions across all the country's 10 provinces. Zimbabwe has a population of around 13 million [20]. The country has a generalised HIV epidemic and prevalence has continued to hover between 13% and 16% for the past decade. Currently there are about 1.4 million PLHIV of which 1.0 million (71%) were on (ART) by December 2018 [18]. Since 2004 the country has adopted all the successive changes in the WHO recommendations to start ART. Patients were eligible to start ART when they had a CD4 count <350 cells/mL in 2010, CD4 <500 cells/mL in 2013 and since 2016, patients are eligible regardless of their CD4 count (Treat All). The country has made significant progress towards achieving the 90-90-90 targets. In a recent survey, 74.2% reported knowing their HIV status; 86.8% self-reported current use of ART and among those who self-reported current use of ART, 86.5% were virally suppressed [21].

During the period between 2011 and 2015, the Zimbabwe National AIDS Strategic Plan II (ZNASP-II) 2011–15 was developed to guide the scale-up of HIV care and treatment services towards universal access [22]. In 2013, the country adopted the 2013 WHO guidelines which recommended the CD4 < 500 cells/mL threshold for ART initiation. A 'test and treat' approach was adopted for all HIV-positive children under 5 years, TB/HIV co-infected, HBV/

HIV co-infected, the HIV-positive partner in HIV sero-discordant relationship and pregnant and breastfeeding mothers (Option B+). The preferred first-line regimen for adults, adolescents, and older children was changed from stavudine, lamivudine and nevirapine (d4T/3TC/NVP) to a once-daily pill of tenofovir, lamivudine and efavirenz (TDF/3TC/EFV) with zidovudine and nevirapine as alternatives for TDF and EFV respectively. After ART initiation stable patients were reviewed every 3 months and drugs were dispensed directly from the health facilities. Monitoring was mainly clinical (weight, WHO clinical stage and assessment of opportunistic infections) at every visit, complemented with laboratory tests (CD4 testing) every 6 months. Compared to the previous period access to CD4 testing improved significantly. By the end of 2015, laboratory based CD4 testing was available in each of the 63 districts as compared to 47 in 2009 [23]. On top of laboratory based CD4 testing, more than 265 point of care (POC) CD4 testing devices were procured and distributed throughout the country mainly to support the roll out of Option B+ [24]. Blood collection and transportation systems were put in place to support facilities without access to CD4 testing. Routine viral load testing was not available in the public sector and there was limited access to targeted viral load testing for patients with suspected treatment failure. The monitoring and evaluation system was mainly paper based. A few high-volume facilities started to implement an electronic patient monitoring system (ePMS) which operated concurrently with the usual paper-based system.

## Sample size and sampling criteria

A minimum sample of 4000 patient charts was required to estimate 12-month ART retention (the outcome of interest) after assuming: 50% ART retention at 12 months after initiation, 20% of charts would be missing, a margin of error of 5% and a design effect of 2.0. The design effect of 2 was used to correct the loss of effective sample which was anticipated from intra-stratum correlation. For logistical and financial feasibility purposes, sampling was restricted to 1,389 ART sites across all 10 provinces that were supporting ≥50 HIV-positive clients on ART for at least 12 months by January 2016. The sites were stratified into seven strata according to their patient volumes receiving ART care and ordered by province and district within each stratum. From these, 70 (5%) ART sites were sampled using a probability proportional to size sampling criteria based on client volumes receiving ART at each site. At the sampled sites, a line-listing was generated in Excel for all HIV positive children, adolescents, adults and pregnant women initiated on ART at the sampled sites between 1 October 2012 and 31 January 2015 using their unique ART numbers obtained from the facility ART register. Following this, the required number of patient ART care booklets were randomly selected without replacement according to their stratum based on random schedule generator in Stata. If a selected ART care booklet was not traceable, the next record was traced until the required sample size by site was reached.

## Study variables and treatment outcomes

The data were collected manually from the patient files. The data collected included demographics (age, sex and marital status), clinical and programmatic variables. Baseline clinical parameters collected at point of ART initiation included WHO stage, functional status and pregnancy. Programmatic data collected included level of care, date of HIV testing, date of enrolment and date of ART initiation. Date of last clinic visit, date of next scheduled clinic visit and date of transfer-out/death/stopping ART were also collected and used to determine the patient status (i.e. active on treatment, LTFU, dead, transferred out and stopped ART). The patient status was determined on the date of data abstraction, regardless of previous treatment interruptions. Patients were considered LTFU if they were more than 180 days without

visit at the clinic on the date of data abstraction. If a patient was reported as LTFU in the patient file, data collectors checked pharmacy refill records to confirm LTFU. Refill dates were considered as visit dates.

### Data collection procedures

Ten teams of three data abstractors collected the data using a structured tool. The data abstractors underwent a 5-day training programme to acquaint them with study tools and study procedures. Data collection tools were piloted to identify weaknesses which then were rectified. The primary source document for the study was the patient manual medical records (Patient OI/ART Booklets) being kept at the health institution. Other sources to complement or validate the information included the different registers and interviews with at least one experienced health care worker at each sampled site. Data was collected using Open Data Kit (ODK) software on Android devices. Data was downloaded at regular intervals for cleaning, quality control checks, merging and backing up. Data from a sample of 10% of the selected charts was re-abstracted as a quality control check measure. Discrepancies were addressed by both the team leader and data abstractors.

### Data analysis

The data were analysed using Stata/IC 14.2 [25]. Our sample size calculation was adjusted to cater for missing data. Patients without outcome data were excluded. Explanatory variables which had more than 30% missingness were also excluded from the analysis. We assumed missingness to be at random. Categorical variables which had less than 30% missingness were included in the analysis (Table 1). We used the missing indicator method, showing the proportion missingness for the different variables (if any). The primary outcome was attrition, and this was defined as either being dead, LTFU or stopped ART.

A multivariable Cox proportional hazard model was used to compute crude and adjusted hazard ratios and their 95% confidence intervals estimating the association between explanatory variables and attrition. The proportional hazards assumption was assessed using log-log plots and Kaplan-Meier versus Cox predicted plots and results of these analyses suggested that the proportional hazards assumption holds. Time to attrition was calculated as the time between the date of ART initiation and the date of outcome or last patient clinic visit. Those active on ART and transferred out were censored on the date of data abstraction and the date they were transferred out respectively. Kaplan-Meier techniques were used to estimate retention on ART at 6, 12, 24 and 36 months. To identify independent risk factors for attrition, first variables were selected into the multivariable model based using a stepwise forward approach (p-value < 0.1). Moreover, sex and level of care were included based on their clinical and biological relevance in HIV retention. Second, stepwise backward elimination was used until all remaining variables were significantly associated with attrition (p-value < 0.05). In an additional analysis we estimated the correlation between year of ART initiation and attrition until 12 months of ART. Patients who started ART in 2015 were excluded since their maximal follow-up time was too short (S1 Table).

### Ethical considerations

The evaluation protocol received ethics approval from Medical Research Council of Zimbabwe (MRCZ/A/2033). The protocol was also sanctioned by the MoHCC. Confidentiality and anonymity of ART clients was protected use their unique ART numbers as no names were abstracted. Permission to conduct the evaluation was sought at various levels of the tiered health delivery system. All data abstractors signed confidentiality forms.

**Table 1. Baseline characteristics of 3810 patients who started ART between 2012–2015, in Zimbabwe.**

|  | N | (%) |
|---|---|---|
| **Total** | 3810 | 100.0 |
| **Year of ART initiation** |  |  |
| 2012 | 289 | (7.6) |
| 2013 | 1476 | (38.7) |
| 2014 | 1850 | (48.6) |
| 2015 | 195 | (5.1) |
| **Sex** |  |  |
| Female | 2262 | (59.4) |
| Male | 1548 | (40.6) |
| **Median Age (N = 3700) (IQR)** | 35 (28–42) |  |
| **Age groups** |  |  |
| Children (0–15 years) | 254 | (6.7) |
| Adolescents & young adults (15-24years) | 385 | (10.1) |
| Adults (25–49 years) | 2735 | (71.8) |
| Elderly (>=50 years) | 424 | (11.1) |
| Missing | 12 | (0.3) |
| **Pregnancy at ART initiation** |  |  |
| Confirmed | 208 | (5.5) |
| Not confirmed | 3602 | (94.5) |
| **Marital status of patient** |  |  |
| Single & divorced | 863 | (22.7) |
| Married | 2125 | (55.8) |
| Widowed | 434 | (11.4) |
| N/A Child | 318 | (8.3) |
| Missing | 70 | (1.8) |
| **Baseline Functional Status** |  |  |
| Impaired | 811 | (21.3) |
| Normal | 2796 | (73.4) |
| Missing | 203 | (5.3) |
| **Baseline WHO Stage** |  |  |
| WHO Stage I | 821 | (21.5) |
| WHO Stage II | 1245 | (32.7) |
| WHO Stage III | 1447 | (38.0) |
| WHO Stage IV | 120 | (3.1) |
| Unknown | 177 | (4.6) |
| **Level of Care** |  |  |
| Central Hospital | 888 | (23.3) |
| Provincial Hospital | 341 | (9.0) |
| Mission/District Hospitals | 1549 | (40.7) |
| Primary Health Care facilities | 1032 | (27.1) |

N—Number of observations, ART—Antiretroviral Therapy, IQR—Interquartile Range, N/A—Not applicable, WHO—World Health Organisation.

## Results

A total of 3,993 (99.8%) records were abstracted out of a target of 4,000. On data cleaning 3,810 (95.4%) records were found to be of quality standard for the analysis (Fig 1).

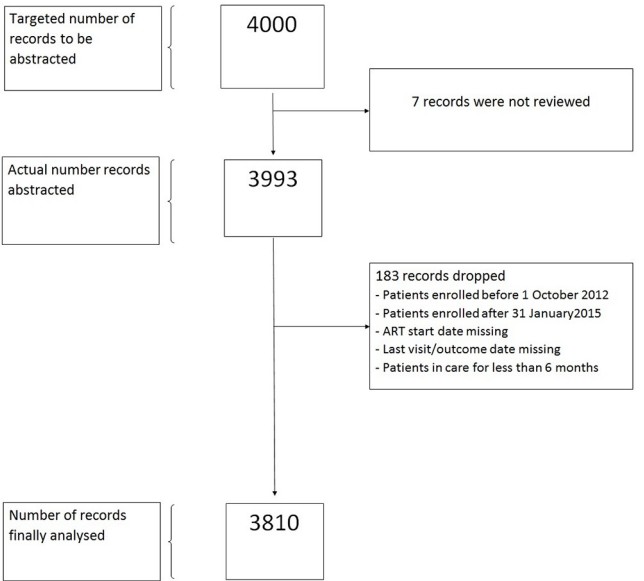

**Fig 1. Study population.**

Of 3,810 patients included, 38.7% and 48.6% started ART in 2013 and 2014, respectively, and 59.4% were female. The median age in years was 35 [interquartile range (IQR):28–42]. Most patients were adults (71.8%), married (55.8%), and had a normal functional status (73.4%). Few (3.1%) were in WHO stage IV, and 27.1% started ART at a primary health care (PHC) facility. At time of ART initiation 5.5% of women were confirmed to be pregnant (Table 1).

Fig 2 shows that the proportion of patients receiving ART at PHC facilities increased gradually from 21.6% in 2012 to 34.5% in 2015.

Overall, retention at 6, 12, 24 and 36 months was 92.4%, 86.5%, 79.2% and 74.4%, respectively. Table 2 shows a comparison between adult patients who started ART between 2007–

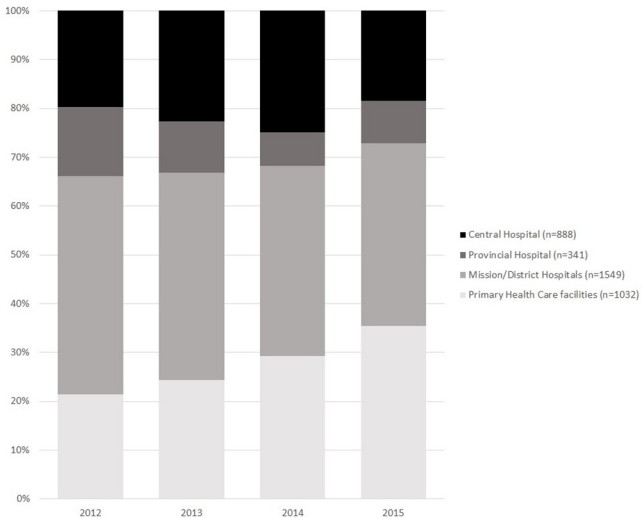

**Fig 2. Level of care where the 3,810 patients started ART between 2012–2015, in Zimbabwe.**

**Table 2. Retention in care of patients who started antiretroviral therapy between 2007–2010 and 2012–2015 in Zimbabwe.**

| | Months since ART initiation | 2012–2015 Evaluation | | | 2007–2010 Evaluation | | | Difference | | |
|---|---|---|---|---|---|---|---|---|---|---|
| | | N | Retention (%) | 95% CI | N | Retention (%) | 95% CI | % | 95%CI | [a]P value |
| Overall | 6 | 3476 | 92.4 | (91.5–93.2) | | | | | | |
| | 12 | 3066 | 86.5 | (85.3–87.5) | | | | | | |
| | 24 | 1427 | 79.2 | (77.7–80.6) | | | | | | |
| | 36 | 222 | 74.4 | (72.5–76.1) | | | | | | |
| Children < 15 years | 6 | 250 | 97.3 | (94.4–98.7) | | | | | | |
| | 12 | 230 | 94.2 | (90.5–96.4) | | | | | | |
| | 24 | 114 | 86.3 | (81.0–90.2) | | | | | | |
| | 36 | 17 | 82.3 | (75.5–87.4) | | | | | | |
| Adults > 15 years | 6 | 3222 | 92.0 | (91.1–92.9) | 3739 | 90.7 | (86.1–93.8) | 1.3 | (-0.02–2.6) | 0.060 |
| | 12 | 2832 | 85.9 | (84.7–87.0) | 3641 | 78.1 | (67.7–84.7) | 7.8 | (5.9–10.0) | <0.001 |
| | 24 | 1311 | 78.7 | (77.1–80.1) | 2003 | 68.8 | (58.5–77.5) | 9.9 | (6.9–12.9) | <0.001 |
| | 36 | 204 | 73.7 | (71.8–75.6) | 806 | 64.4 | (55.7–72.3) | 9.3 | (2.4–16.2) | <0.001 |

[a]: Two-sample test of proportions

2010 and 2012–2015. Retention at 6, 12, 24 and 36 months was 1.3%, 7.8%, 9.9% and 9.3% higher among patients who started between 2012–2015. Differences were statistically significant except at 6 months.

The maximum follow-up time for the patients who started ART in 2012, 2013, 2014 and 2015 was 40.4, 37.2, 25.2 and 13.4 months respectively (S1 Table). We assessed 12-month retention by year of ART initiation with those who started ART in 2015 excluded. Retention decreased during the study period. Those who started ART in 2013 and 2014, as compared to those who started ART in 2012, were 2.18 and 3.78 times at risk of attrition at 95% confidence interval (CI), respectively (Fig 3).

Over 6489 years of patient follow-up; 77.4% were active on treatment, 2.4% transferred out and 20.2% were lost through attrition (LTFU 19.8%, and died 0.4%) The median follow-up time per patient was 1.7 years (IQR 1.1–2.4). The overall attrition rate was 11.8, [95% confidence interval (CI):11.0–12.7] per 100 person years (PY). The other attrition rates are shown in Table 3.

Variables which had p -value < 0.1 (age group, marital status, year of ART initiation, WHO stage, functional status and pregnancy) were included in multivariable analysis. Sex and level of care were included based on their clinical and biological relevance in HIV retention. In multivariable analysis, being an adolescent or a young adult [multivariable-adjusted hazard ratio (aHR)(vs adult; 1.41; 95% CI:1.14–1.74], receiving care at PHC facility and district level (vs central and provincial facility; aHR 1.23; 95% CI:1.01–1.49) and aHR 1.21; 95% CI:1.01–1.44), having initiated ART between 2014–2015 (vs 2012–2013; aHR 1.45; 95% CI:1.24–1.69), having WHO Stage 4 (vs Stage I–III; aHR 2.06; 95% CI:1.51–2.81) and having an impaired functional status (vs normal status; aHR1.25; 95% CI:1.04–1.49) were associated with increased attrition in our setting. Risk of attrition among children (<15years) in our setting was about 40% lower (vs adults; aHR 0.64; 95% CI:0.46–0.91 (Table 3).

## Discussion

This follow-up evaluation of the Zimbabwe National ART programme, which included children, adolescents and adults started on ART between 2012 and 2015, showed an overall improvement of the performance of the ART programme compared to those started on ART

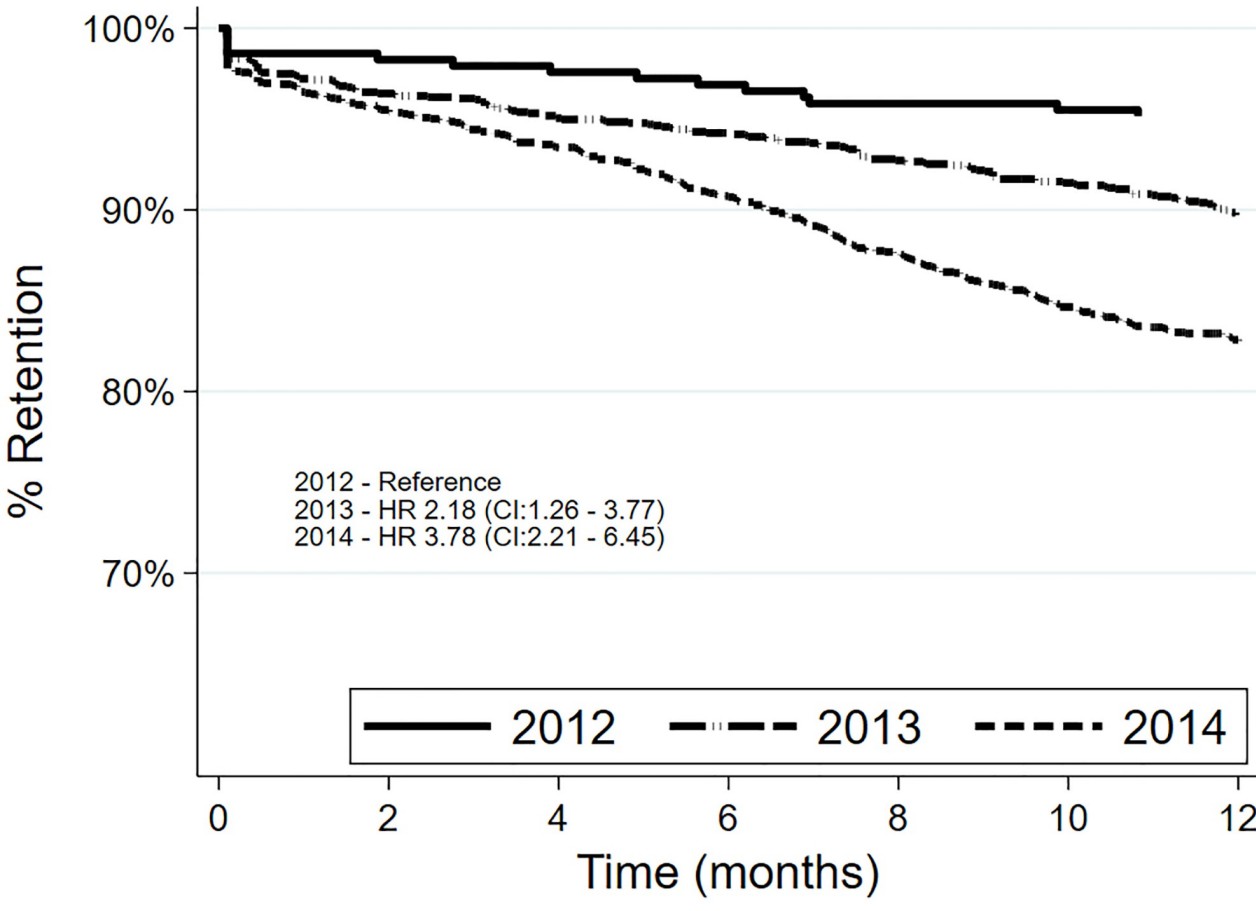

**Fig 3. Comparison of retention rates of patients who started antiretroviral therapy in 2012, 2013 and 2014. in Zimbabwe.**

between 2007–2010 [17]. Attrition was mainly explained by LTFU. Retention observed in this evaluation was above what is reported for children, adolescents and adults in other similar studies from low-resource settings [26–28]. Adolescents and young adults, patients with advanced HIV disease (WHO Stage 4, impaired functional status), those receiving care at PHC level and starting treatment after 2013, when the country switched treatment guidelines to the 500 CD4 cells/mL threshold, were at risk of attrition.

The increase in retention since the previous evaluation may be explained by strategies put in place by the MoHCC. Relying on a vast body of evidence showing that decentralization increases access to ART and retention in care [29], the prior evaluation recommended further decentralization. Moreover, other innovative strategies to improve patient retention, such as adherence clubs, food supplementation and mobile short messages service (SMS) reminders were recommended [17]. Our data show a substantial increase of the proportion of patients receiving ART at PHC level. This decentralization was supported by policy shifts (in particular, the provision for nurses to initiate ART in non-complicated cases), and by significant investments in training, supportive supervision and clinical mentoring [30]. In addition to decentralization, the improvement in retention could also be attributed to adoption of WHO guidance for earlier ART initiation among those tested HIV-positive with CD4<500 cells/mL compared to the previous 350 cells/mL CD4 threshold. For instance in this cohort 41.1% had WHO stage 3 or 4 compared to 87.6% in the previous study [17]. However, other

**Table 3. Bivariate and multivariate analysis of factors associated with attrition in the 2012–2015 ART cohort.**

| Variable | Categories | Total | PT years | Attrition$ | Attrition % | Attrition Per 100PY | HR | p | 95%CI | aHR | P | 95%CI |
|---|---|---|---|---|---|---|---|---|---|---|---|---|
| **Total** | | **3810** | **6489** | **768** | 20.2 | **11.8** | | | | | | |
| Sex | Female | 2262 | 3804 | 440 | 19.5 | 11.6 | 1 | | | | | |
| | Male | 1548 | 2685 | 328 | 21.2 | 12.2 | 1.07 | 0.377 | (0.92–1.23) | | NS | NS |
| Age group | Adults | 2735 | 4650 | 555 | 20.3 | 11.9 | 1 | | | 1 | | |
| | Children | 254 | 467 | 35 | 13.8 | 7.5 | **0.64** | **0.009** | **(0.45–0.89)** | **0.64** | **0.011** | **(0.46–0.91)** |
| | Adolescents & young adults | 385 | 562 | 103 | 26.8 | 18.3 | **1.47** | **<0.001** | **(1.19–1.82)** | **1.41** | **0.002** | **(1.14–1.74)** |
| | Elderly | 424 | 784 | 75 | 17.7 | 9.6 | 0.81 | 0.096 | (0.64–1.04) | 0.83 | 0.123 | (0.65–1.05) |
| | Missing | 12 | | | | | | | | | | |
| Marital Status | Married | 2125 | 3619 | 419 | 19.7 | 11.6 | 1 | | | | | |
| | Single & divorced | 863 | 1382 | 202 | 23.4 | 14.6 | **1.25** | **0.010** | **(1.05–1.47)** | | | |
| | Widowed | 434 | 772 | 85 | 19.6 | 11.0 | 0.97 | 0.766 | (0.76–1.22) | | NS | NS |
| | Child N/A | 318 | 584 | 49 | 15.4 | 8.4 | 0.74 | 0.042 | (0.55–0.99) | | | |
| | Missing | 70 | | | | | | | | | | |
| Level of Care | Central & province | 1229 | 2119 | 230 | 18.7 | 10.9 | 1 | | | 1 | | |
| | Primary health care facility | 1032 | 1658 | 213 | 20.6 | 12.8 | 1.15 | 0.134 | (0.96–1.39) | **1.23** | **0.039** | **(1.01–1.49)** |
| | District | 1549 | 2712 | 325 | 21.0 | 12.0 | 1.11 | 0.242 | (0.93–1.31) | **1.21** | **0.032** | **(1.01–1.44)** |
| Year ART Initiation | 2012–13 | 1765 | 3994 | 377 | 21.4 | 9.4 | 1 | | | 1 | | |
| | 2014–2015 | 2045 | 2495 | 391 | 19.1 | 15.7 | **1.45** | **<0.001** | **(1.24–1.69)** | **1.45** | **<0.001** | **(1.24–1.69)** |
| WHO Stage | I-III | 3513 | 5944 | 675 | 19.2 | 11.4 | 1 | | | 1 | | |
| | IV | 120 | 194 | 43 | 35.8 | 22.2 | **1.95** | **<0.001** | **(1.43–2.66)** | **2.06** | **<0.001** | **(1.51–2.81)** |
| | Missing | 177 | | | | | | | | | | |
| Functional status | Normal | 2796 | 4774 | 538 | 19.2 | 11.3 | 1 | | | 1 | | |
| | Impaired | 811 | 1320 | 177 | 21.8 | 13.4 | 1.18 | 0.058 | (0.99–1.40) | **1.25** | **0.013** | **(1.04–1.49)** |
| | Missing | 203 | | | | | | | | | | |
| Pregnant when starting ART | Not confirmed | 3602 | 6219 | 718 | 19.9 | 11.5 | 1 | | | | | |
| | Confirmed | 208 | 270 | 50 | 24.0 | 18.5 | **1.48** | **0.007** | **(1.12–1.98)** | | NS | NS |

NS = Not significant (not included in final model); HR = Hazard Ratio; aHR = adjusted Hazard Ratio; CI = Confidence Interval; ART = Antiretroviral Therapy;
WHO = World Health Organisation
$ either death or LTFU

recommended measures, such as food supplementation, adherence clubs, mobile short messages service (SMS) reminders, fast tracking of stable patients on ART, community and family ART refill groups, were not implemented on a wide scale in the public sector.

However, the increase of decentralized ART does probably not explain the higher level of retention, compared to the previous evaluation. Surprisingly, our study showed that receiving ART at PHC level was associated with attrition. This finding contrasts with the findings of the previous evaluation. In the previous evaluation, retention among patients receiving ART at PHC level was better than among those initiating ART at higher levels of care, in particular district/mission hospitals [17]. We speculate that the higher level of attrition in PHC facilities may be explained by the massive decentralization (down referral), which happened during the current evaluation period, whereby patients who started ART at district/provincial hospitals were referred to PHC level for follow-up. Implementation of policy changes may be abrupt and have an adverse effect. After assessment by a healthcare worker, stable patients may have little or no opportunity to object against the decision to be referred to another clinic. Patients referred to a clinic which is not of their choice are more likely to self-transfer to another facility of their choice. Such patients are then considered as LTFU in one clinic, while retained in another clinic. Moreover, at the beginning of the decentralization process, care at PHC level may not have been fully developed. Another study showed that abrupt down-referral may lead

to a decrease in quality of care and resulting in worse health outcomes [31]. When PHC facilities are ill prepared, usually outcome monitoring is poor. The country has almost completed the decentralization process for ART services and the focus now should be on improving the quality of care at PHC facilities. The necessary health system support structures which include human resources for health (recruitment, training and capacity building), drug supply, monitoring and evaluation should be prioritised.

Most of the reported attrition (98%) was due to LTFU. This finding was similar to the prior evaluation where LTFU also accounted for the larger proportion of attrition. Determining true LTFU is not always easy. A recent meta-analysis showed that, of patients LTFU and traced, 30% had self-transferred, 30% had stopped taking ART and the other 30% had died [32]. We therefore hypothesize that the increased LTFU in our study might be administrative, especially at PHC level where monitoring is less well developed. A substantial proportion of patients may seem LTFU, but be in care at another or even the same clinic with another identification number [31]. The probability of administrative reasons for LTFU is probably higher at PHC facilities, especially when paper-based tools are used to monitor and report treatment outcomes [31]. Poor documentation of clinic visits and transfers in medical records may result in administrative LTFU. There is need to determine the true nature of the outcome of patients LTFU within the Zimbabwe ART programme. A substantial proportion may be alive and receiving ART from another facility, while others may have died but were not reported as such [33,34]. This may be achieved through strengthening the current active patient tracking and tracing mechanisms [35]. The existing ART programme's electronic patient monitoring systems (ePMS) should also be scaled up and optimised to bring efficiency in patient monitoring [36]. Tracing mechanisms include SMS reminders, phone calls and home visits by community health workers [37].

There was a change in the risk factors for attrition in comparison to the previous evaluation. Males were at risk of attrition in the prior evaluation but were not at risk in the current evaluation. Several studies have consistently showed males at high risk of attrition due to several reasons, including employment related constraints and poor health seeking behaviour leading to late presentation [13,15,38,39]. During the study period the country did not implement many specific interventions that targeted men. Decentralization of ART services may have also worked in favour of males for better retention as they could now access services at health facilities closer to where they reside. Another strategy that may have improved male ART uptake and retention was male involvement in PMTCT (Option B+), promoted during massive campaigns [19]. There is a need for further studies to assess whether this strategy suffices, or if other strategies targeting men are needed, such as flexible clinic hours to accommodate work, community-based ART delivery, and tracing of those who miss appointments [40,41].

The previous evaluation did not include data on children (< 15 years) but included adolescents (>=15 years) and young adults. Though age was included the previous evaluation did not assess the effect of age on attrition [17]. In our study we found children (< 15 years) to be at lower risk of attrition as compared to adults. Our findings seem to contrast with those from previous studies, which identified dependency of guardians, lack of palatable formulations, and challenges with disclosure as additional challenges experienced by children [42–44]. We speculate that the following interventions impacted positively on retention in our setting: PMTCT Option B+ targeting the mother-baby pair, integration of paediatric ART services with immunisation, decentralization of paediatric ART services, improved access to palatable paediatric ART formulations, and emphasis on secondary caregivers to support the primary caregivers with the child's adherence in case of absence or other commitments [45–48].

We found that adolescents and young adults were more at risk of attrition when compared to adults. Our findings are consistent with other studies [49–52]. Adolescents and young adults

have been shown to be at high risk of attrition due to several factors, which include lack of youth-friendly services, rigid scheduling not taking into account schooling, and unavailability of peer caregivers [53,54]. Addressing these challenges can lead to improvement in the retention of adolescents and young adults on ART. Locally, community adolescent treatment supporters (CATS) have been shown to improve retention among adolescents and scaling up of the initiative should be prioritised [55].

We found patients with advanced disease to be at higher risk of attrition. Patients with advanced HIV disease are prone to attrition mainly due to mortality and morbidity. The common causes of morbidity and mortality in low resource setting include cryptococcal meningitis, tuberculosis, sepsis, malignancy and wasting syndrome/chronic diarrhoea [56–58]. Advanced HIV disease was also a risk factor for attrition in the previous evaluation. Very minimal investment in building capacity at primary health facilities in the management of patients with advanced HIV disease contributes too poorer outcomes in this subgroup. Indeed, access to baseline CD4 testing, screening and management of opportunistic (tuberculosis, cryptococcal meningitis) and access to prophylaxis (isoniazid preventive therapy & pre-emptive fluconazole) remains a challenge in the Zimbabwe's public sector [59,60]. Recent evidence has also shown that screening, prophylaxis and management of opportunistic diseases (mainly tuberculosis, bacterial sepsis and cryptococcal disease) significantly reduce morbidity and mortality [61,62]. However, evidence from this enhanced care package is currently being poorly implemented by most ART programmes in low resource setting [63]. There is a need to mobilise resources for training and capacity building of health workers, setting up the necessary infrastructure and procurement of the necessary commodities required in the management of patients with advanced disease. However, the process should be guided by a formal assessment of the current burden of advanced disease in the country.

We also found that patients who started ART in 2013 or later were at risk of attrition as compared to those who started in 2012. We speculate that this could be explained by decentralization and the fact that patients were now starting ART at a higher CD4 count. Decentralization happened gradually between 2012–2015. By the end of 2015 more patients were receiving care at PHCs, and our study found attrition to be higher at PHCs. The number of patients who were being initiated and followed-up on ART at PHCs increased during the period (Fig 1). We speculate that the gradual increase of patients starting ART at the possibly ill-prepared PHCs might have contributed to the increase in attrition over time during the study period. In 2013, the country switched ART guidelines to the 500 CD4 cells per cubic millilitres threshold from the 350 cells per cubic millilitres cut-off which was used in 2012. On top of the change in the CD4 cut-off point, test and treat was also introduced to specific sub-populations (children under 5 years, TB/HIV co-infected, HBV/HIV co-infected, the HIV-positive partner in HIV sero-discordant relationship and pregnant and breastfeeding mothers (Option B+). Similar findings were reported in previous studies, which showed that patients who started ART at a higher CD4 were at risk of attrition [8,9]. Patients starting ART at high CD4 are less sick and have low risk perception which might affect adherence to long term therapy [64]. This then calls for earnest implementation of the current country operational guidelines which recommends adequate psychosocial preparation and readiness assessment before ART initiation under Treat All [37].

Our study was a follow-up evaluation of the Zimbabwe ART programme. To our knowledge, no studies showing national data have been followed-up by a second study of similar magnitude. In a recent review, none of the studies included was a follow-up to a prior evaluation [5]. Despite the highlighted strengths our study had limitations. Most of the reported attrition was due to LTFU. Patients who are classified as LTFU might have been alive and still on ART (self-transferred), stopped ART or may have died [32]. Our LTFU definition

(180 without visit) differed from the one used by the program (LTFU = 90 days late), due to data availability. However, considering that patients received up to 3 months ART refill, both definitions would result in similar findings. We could not report on immunologic and virologic patient outcomes due to missing information in the patient manual medical records. Information on viral load testing and CD4 testing was more than 30% missing. These variables were excluded. To preserve all observations for variables with less than 30% missingness the missing indicator method was used. We assumed that missingness was at random, and mainly explained by health workers inconsistently completing the medical records. However, the method can affect the residual variance of the regression [65]. Moreover, we did not assess differences between clinics of the same level of care.

## Conclusion

Zimbabwe's ART program shows good retention across different patient groups. The ART retention increased since the previous 2011 evaluation. However, adolescents and young adults, patients with advanced HIV disease (WHO Stage 4, impaired functional status), receiving care at PHC level and starting treatment after 2013 when the country switched treatment guidelines to the 500 CD4 cells per cubic millilitres threshold were at risk of attrition.

From the findings we recommend research into the following areas: reasons for higher attrition at PHC facilities, feasibility of screening for advanced disease at primary health facilities and ensuring access to referral clinical care, assessing retention within levels of care, qualitative research to explore the causal link between attrition in recent years, and being less sick at the start of treatment. To improve monitoring electronic patient monitoring systems should be prioritised to aid patient tracking. There is also a need for differentiated care strategies for adolescents and young adults to improve retention. Creation of youth-friendly services, flexible scheduling of visits and expansion of peer caregivers/treatment supporters should be considered.

## Supporting information

**S1 File. Patient Retention, Clinical Outcomes and Attrition-Associated Factors of HIV-Infected Patients Enrolled in Zimbabwe's National Antiretroviral Therapy Programme, 2007–2010.**
(PDF)

**S1 Table. Follow-up time of the 3810 patients who started ART in Zimbabwe between 2012–2015.**
(DOCX)

## Acknowledgments

The authors of the articles would like express gratitude and appreciation to the following organizations that made immense contributions and technical support to the success of this study namely the Ministry of Health and Child care; National AIDS Council of Zimbabwe (NAC); University of Zimbabwe, College of Health Sciences (UZCHS), Department of Community Medicine; World Health Organization (WHO) and the Institute of Tropical Medicine (ITM), Antwerp, Belgium. The authors would like to appreciate those who participated in training, data abstraction, data cleaning, data entry and regular progress and review meetings during the study: Gerald Shambira, Rosemary Mhlanga-Gunda, Mutsa Mhangara, Angela Mushavi, Amon Mpofu, Brilliant Nkomo, Ngwarai Sithole, Isaac Taramusi, Joseph Murungu, Solomon Mukungunugwa and Christine C. Chakanyuka-Musanhu. We would also like to thank Bart

Karl Jacobs from the Institute of Tropical Medicine, Antwerp Belgium for statistical review and providing feedback on drafts of the paper. We are particularly indebted to the MoHCC management at different levels for the support and permission to enter the respective sampled facilities. Finally, we thank the data abstraction teams, Open Data Kit support team, front line health workers and people living with HIV on ART whose records were reviewed.

## Author Contributions

**Conceptualization:** Richard Makurumidze, Tsitsi Mutasa-Apollo, Tom Decroo, Regis C. Choto, Kudakwashe C. Takarinda, Janet Dzangare, Lutgarde Lynen, Wim Van Damme, James Hakim, Simbarashe Rusakaniko.

**Data curation:** Richard Makurumidze, Tsitsi Mutasa-Apollo, Tom Decroo, Kudakwashe C. Takarinda, Janet Dzangare, Simbarashe Rusakaniko.

**Formal analysis:** Richard Makurumidze, Tom Decroo, Simbarashe Rusakaniko.

**Funding acquisition:** Tapuwa Magure, Owen Mugurungi.

**Methodology:** Richard Makurumidze, Tsitsi Mutasa-Apollo, Tom Decroo, Regis C. Choto, Kudakwashe C. Takarinda, Janet Dzangare, Lutgarde Lynen, Wim Van Damme, James Hakim, Simbarashe Rusakaniko.

**Project administration:** Tsitsi Mutasa-Apollo, Regis C. Choto, Tapuwa Magure, Owen Mugurungi.

**Resources:** Tapuwa Magure, Owen Mugurungi.

**Supervision:** Tsitsi Mutasa-Apollo, Regis C. Choto, Tapuwa Magure, Owen Mugurungi, Simbarashe Rusakaniko.

**Validation:** Richard Makurumidze, Tom Decroo, Simbarashe Rusakaniko.

**Writing – original draft:** Richard Makurumidze.

**Writing – review & editing:** Tsitsi Mutasa-Apollo, Tom Decroo, Regis C. Choto, Kudakwashe C. Takarinda, Janet Dzangare, Lutgarde Lynen, Wim Van Damme, James Hakim, Simbarashe Rusakaniko.

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
