## [Decision Letter · Decision Letter 0]

22 Oct 2019

PONE-D-19-23547

Retention and predictors of attrition among patients who started antiretroviral therapy in Zimbabwe’s National Antiretroviral Therapy Programme between 2012 and 2015

PLOS ONE

Dear Dr Makurumidze,

Thank you for submitting your manuscript to PLOS ONE. After careful consideration, we feel that it has merit but does not fully meet PLOS ONE’s publication criteria as it currently stands. Therefore, we invite you to submit a revised version of the manuscript that addresses the points raised during the review process.

We would appreciate receiving your revised manuscript by 30th November 2019. To enhance the reproducibility of your results, we recommend that if applicable you deposit your laboratory protocols in protocols.io, where a protocol can be assigned its own identifier (DOI) such that it can be cited independently in the future. For instructions see: http://journals.plos.org/plosone/s/submission-guidelines#loc-laboratory-protocols

We look forward to receiving your revised manuscript.

Kind regards,

Kwasi Torpey, MD PhD MPH

Academic Editor

PLOS ONE

Journal Requirements:

Additional Editor Comments (if provided):

Reviewers' comments:

Reviewer's Responses to Questions

**Comments to the Author**

1. Is the manuscript technically sound, and do the data support the conclusions?

Reviewer #1: Yes

Reviewer #2: Yes

2. Has the statistical analysis been performed appropriately and rigorously? 

Reviewer #1: No

Reviewer #2: Yes

3. Have the authors made all data underlying the findings in their manuscript fully available?

Reviewer #1: Yes

Reviewer #2: Yes

4. Is the manuscript presented in an intelligible fashion and written in standard English?

Reviewer #1: Yes

Reviewer #2: Yes

5. Review Comments to the Author

Reviewer #1: Review

Retention and predictors of attrition among patients who started antiretroviral therapy in Zimbabwe’s National Antiretroviral Therapy Programme between 2012 and 2015

Dear authors,

thank you for submitting this interesting paper on factors influencing retention of the ART program in Zimbabwe.

I do have the following questions and remarks regarding your methodology and interpretation:

Sample size:

Your assumptions regarding the sample size does not include the comparison between the retention rates at 12 months between the actual and the former study.

Reading the section it seems you need 4000 people to estimate a single rate – but then you should maybe give the broadness of a confidence interval to motivate this sample size.

What do you mean by design effect 2.0?

Sampling criteria:

You sampled from 1.10.2012 to 31.12.2015. The analysis was done in 2016?

When I see this information I conclude that in the sub-cohort of those persons starting the ART-Program in 2014/15 there is nobody that can have a retention until 36 months (if starting 2014) and 24 months (if starting in 2015). I don’t find any information at the minimum/maximum length of observation in the (sub-) cohort, could you please add this.

Statistical analysis:

Line 203-5

“Time to attrition was calculated as the time between the date of ART initiation and the date of outcome or last patient clinic visit. Those active on ART and transferred out were censored on the date of data abstraction and the date they were transferred out.”

I don’t understand the second sentence. How can you be censored to two dates?

- As you don’t have information on the time of censoring you could think of using the AFT-Model (Accelerated-Failure-Time-Modell) instead of the Cox-Model. In the AFT-Model you don’t need a date but a period in which an event/censoring occurs.

- Did you check for the proportional hazards assumption of the Cox-Model?

- I suggest to look into subgroups of patients conditioning on the time having been in the ART-program:

e.g. the retention in those patients already having been in ART for 6 months/for 12 months. Maybe the risk of attritions is time dependent and decreases over time.

- Fig 3:

In the method section you describe an hierarchical variable selection for a multivariable model. I would have preferred a more knowledge-based model building – if at all. Could you please mention all the variables selected? I guess the reader is supposed to find out in Fig 3 – but I find bivariate p-values <0.1 and no effect estimate of this variable in the multivariable section – for example in the case of marital status. (what does the N.S. – non significant mean in this situation then?).

Acutally I don’t think you need the multivariable analysis as you want to describe risk groups for higher attrition. The risk groups are well described already in the bivariable analysis. A multivariable model can be used as a prediction model – but this is not what you wanted or what is wanted in dealing with ART-patients, I guess.

In the results section you discuss the single risk factors in comparison to the former analysis – you don’t refer to the a prediction.

- Fig 2 and Fig 3

Please refrain from reporting the p-values besides your main effect (please see the paper ‘Scientists rise up against statistical significance’ in Nature 2019 https://www.nature.com/articles/d41586-019-00857-9)

Fig 3:

- In the sub group 2012/2013 there is much more PT compared to the patients included in 2014/2015 yielding in lower attrition rate/PT.

I think this is because your sampling until 12/2015 limits the potential observation time differentially and therefore fewer persons can be present at 12 months or later in the later cohort. How can somebody in the later subcohort be observed for 36 months? The observation time for somebody starting in 2015 is <12 months.

- You state that the difference in attrition rate between patients starting in the two periods is because of the changing criteria for ART-therapy regarding the cell count. in the introduction and conclusion you write that the change in this criterium happened in 2013.

According to Fig 1 most of the patients in the cohort 2012/2013 should have had the same conditions for inclusion into the program than during 2014/2015.

Other:

- Maybe this paper could be of interest for you as well:

Alhaj M et al. Retention on antiretroviral therapy during Universal Test and Treat implementation in Zomba district, Malawi: a retrospective cohort study. J Int AIDS Soc 2019 Feb;22(2):e25239

- Please check the syntax of the sentences in lines 283 and 390

- Loss-To-Follow-Up not abbreviated correctly all the time – please check!!

Reviewer #2: This is a large retrospective study evaluating the risk of LTFU among individuals who started ART between 2012-2015 in Zimbabwe. We commend the efforts of working with program data to produce this manuscript and support it’s publication.

I believe this study will benefit from the following comments:-

1. The authors should be more specific about the purpose of this study, the comparisons and the hypotheses.

2. How would patients who were considered LTFU (>180 days since without a visit) but show up at the clinic over 6 months later were treated in the analysis? I.e. are there some cases that were LFTU based on this criteria but showed up after 6 months has lapsed?

3. Could the other program activities (Line 294) explain lower retention in the 2014-2015 cohort compared to 2012-2013? (Fig 3). Did these activities diminish over time? I recall reading that more individuals initiated ART over time so does it mean that although ART was scaled up, the retention was lower. Please correct my misinterpretations and kindly explain these results.

4. Discuss program adherence and retention interventions (if any) and consider citing a paper by Coker et al., Curr HIV/AIDS Reports 2015. Socio-Demographic and Adherence Factors Associated with Viral Load Suppression in HIV-Infected Adults Initiating Therapy in Northern Nigeria: A Randomized Controlled Trial of a Peer Support Intervention that highlights that indicate that ART adherence will improve significantly regardless of whether HIV-infected adults received adherence interventions or standard of care services.

Please see attachment for additional comments that could not fit in here.

6. PLOS authors have the option to publish the peer review history of their article (what does this mean?). If published, this will include your full peer review and any attached files.

Reviewer #1: No

Reviewer #2: Yes: Modupe Coker

---

## [Author Response · Author response to Decision Letter 0]

7 Dec 2019

Dear Editor,

Many thanks for the review of “Retention and predictors of attrition among patients who started antiretroviral therapy in Zimbabwe’s National Antiretroviral Therapy Programme between 2012 and 2015". We have carefully read and discussed the recommendations made by the reviewers, and have adapted the paper accordingly.

A marked-up copy of the manuscript highlighting changes made to the original version labeled 'Revised Manuscript with Track Changes' and a clean version labeled 'Manuscript' have been uploaded.

In this response letter we respond to each point raised by the reviewers. Please find our responses below with revisions or additions to the manuscript in quotes. We shall of course be available to respond to any additional question you may have.

Kind regards,

Richard Makurumidze, on behalf of the co-authors

---

## [Editor Report · Decision Letter 1]

18 Dec 2019

Retention and predictors of attrition among patients who started antiretroviral therapy in Zimbabwe’s National Antiretroviral Therapy Programme between 2012 and 2015

PONE-D-19-23547R1

Dear Dr. Makurumidze,

We are pleased to inform you that your manuscript has been judged scientifically suitable for publication and will be formally accepted for publication once it complies with all outstanding technical requirements.

With kind regards,

Kwasi Torpey, MD PhD MPH

Academic Editor

PLOS ONE
---

## [Editor Report · Acceptance letter]

23 Dec 2019

PONE-D-19-23547R1 

Retention and predictors of attrition among patients who started antiretroviral therapy in Zimbabwe’s National Antiretroviral Therapy Programme between 2012 and 2015 

Dear Dr. Makurumidze:

I am pleased to inform you that your manuscript has been deemed suitable for publication in PLOS ONE. Congratulations! Your manuscript is now with our production department. 

With kind regards,

on behalf of

Professor Kwasi Torpey 

Academic Editor

PLOS ONE